# Looking Back on Learned Experiences For Class/task Incremental Learning

**Mozhgan PourKeshavarz**[1]    **Guoying Zhao**[2]    **Mohammad Sabokrou**[1]

[1]School of Computer Science, Institute for Research in Fundamental Sciences (IPM)
[2]Center for Machine Vision and Signal Analysis, University of Oulu, Finland
{pourkeshavarz,sabokro}@ipm.ir   guoying.zhao@oulu.fi

## Abstract

Classical deep neural networks are limited in their ability to learn from emerging streams of training data. When trained sequentially on new or evolving tasks, their performance degrades sharply, making them inappropriate in real-world use cases. Existing methods tackle it by either storing old data samples or only updating a parameter set of deep neural networks, which, however, demands a large memory budget or spoils the flexibility of models to learn the incremented task distribution. In this paper, we shed light on an on-call transfer set to provide past experiences whenever a new task arises in the data stream. In particular, we propose a Cost-Free Incremental Learning (CF-IL) not only to replay past experiences the model has learned but also to perform this in a cost free manner. Towards this end, we introduced a memory recovery paradigm in which we query the network to synthesize past exemplars whenever a new task emerges. Thus, our method needs no extra memory for data buffering or network growing, besides calls the proposed memory recovery paradigm to provide past exemplars, named a transfer set in order to mitigate catastrophically forgetting the former tasks in the Incremental Learning (IL) setup. Moreover, in contrast with recently proposed methods, the suggested paradigm does not desire a parallel architecture since it only relies on the learner network. Compared to the state-of-the-art data techniques without buffering past data samples, CF-IL demonstrates significantly better performance on the well-known datasets whether a task oracle is available in test time (Task-IL) or not (Class-IL)[1].

## 1 Introduction

Conventional Deep Neural Networks (DNNs) are typically trained in an offline batch setting where all data are available at once. However, in real-world settings, models may incrementally encounter new classes when trained online. In such scenarios, deep learning models suffer from catastrophic forgetting McCloskey & Cohen (1989), meaning they forget the previously obtained knowledge when adapting to the new information from the incoming observations. This is mainly because, models overwrite the decisive parameters for earlier tasks while learning the new one.

Incremental Learning (IL) methods aim at training a single DNN from an unlimited stream of data, mitigating catastrophic forgetting conditioned on the limited computational overhead and memory budget. In the IL literature, the typical setting is a model to learn many tasks sequentially where each task contains one or more disjoint classification problems. The majority of the existing works assume that task identities are provided at the test time so that one can select the relevant part of the network for each example. This setup has been named task-IL, where a more general circumstance is that task labels are available only during training, has been named class-IL.

Recently, various approaches have been put forward to solve catastrophic forgetting. Some works investigate on parameter isolation approaches using either expanding network architecture (Aljundi et al., 2017; Xu & Zhu, 2018; Yoon et al., 2017; Rosenfeld & Tsotsos, 2018) or considering a fixed-sized set of model parameters (Masse et al., 2018), pruning (Mallya & Lazebnik, 2018) to

---

[1]The code is available at https://github.com/MozhganPourKeshavarz/Cost-Free-Incremental-Learning

learn a new task. Thus, they are only applicable to the task incremental scenario. Besides, several studies suggest a regularization term added to the loss function (Kirkpatrick et al., 2017; Aljundi et al., 2018; Zenke et al., 2017; Nguyen et al., 2017) to preserve the knowledge of previous classes without a need for knowing the task label during inference. Despite the increasing number of these methodologies, they can not reach a reasonable performance. Another followed terminology is experience replay-based methods which store actual data samples from the past (Buzzega et al., 2020; Riemer et al., 2018; Chaudhry et al., 2018) to alleviate catastrophic forgetting problem, where more of the recent state-of-the-art (SOTA) works fall into this category. Despite the success of this approach, they suffer from a main limitation that is the fixed memory budget when increasing the number of classes, leading to severe performance degradation. To overcome this constraint, several works propose to generate past data samples (Zhai et al., 2019), as pseudo-data examples, conditioned on the model underlying distribution. Although their method are considered a data-free approach by discarding the need of storing real data, they achieve better performance than previous data-free works. However, one main drawback that these approaches encounter is a need for a high-capacity auxiliary network to generate data which increase the complexity of the system in terms of training-difficulty and computation overhead.

In this paper, we propose a novel Cost-Free Incremental Learning (CF-IL) paradigm, which allows the model to remember previous knowledge by itself, thereby eliminating the need for external memory. Also, in the proposed method, there is no need for a parallel network to generate past exemplars. Besides, we ask the learner network to remember what it has learned before learning new classes. Inspired by (Fredrikson et al., 2015; He et al., 2019; Nayak et al., 2019), we propose a solution to reply the previous experiences by recovering the learner network memory. The main idea is having only the learned network to not only remember what it has learned but also stay learned while encountering newly received knowledge, much like how humans continually learn and remember them by themselves with no external help. The most likely method to learn in this way is the regularization-based method. However, they still could not compromise between human-like learning and performance. Besides, our method can achieve a better result by a significant margin with no additional cost.

In summary, the main contributions of this paper are: (1)To the best of our knowledge, we introduce, for the first time, the idea of CF-IL to make experience replay without a need to a parallel auxiliary network, or any other additional requirement, (2)Since we only rely on the learner network, there is no need for buffering past data samples, extracted prior knowledge, and a parallel high-capacity network, resulting in significantly reduced computational cost and memory footprint, and (3)Our method compromises between decreasing memory budget and maintaining performance comparing to the current SOTA task on both class and task incremental learning methods.

## 2 RELATED WORK

Several studies have been carried out to tackle catastrophic forgetting which falls into four categories as follows.

***Model Growing:*** This approach is the first investigations into incremental learning in which a new set of network parameters is dedicated for each task in a dynamic architecture networks (Aljundi et al., 2017; Xu & Zhu, 2018; Yoon et al., 2017; Rosenfeld & Tsotsos, 2018). Therefore, they require the task label to be known at test time to trigger the correct state of the network, which can not be employed in the class-IL setting where task labels are not provided. Moreover, maintaining class incremental performance can be met at the cost of limited scalability.

***Memory Replay:*** These methods allow access to a fixed memory buffer to hold actual samples (Rebuffi et al., 2017; Isele & Cosgun, 2018; Chaudhry et al., 2019; Rolnick et al., 2018). When learning new tasks, samples from the former tasks are either reused as model inputs for rehearsal or to constrain optimization (Lopez-Paz & Ranzato, 2017; Chaudhry et al., 2018) of the recent task loss to prevent preceding task interference. Although this approach is pioneering, its main limitation is a fixed budget, which caused the bias problem (Hou et al., 2019; Wu et al., 2019) by the imbalanced number of old and new class training samples (Belouadah & Popescu, 2019). In the meantime, some works explore a pseudo-memory replay (Robins, 1995; Atkinson et al.) approach in which a generative model is adopted to synthesize past samples when facing a new task resulting increasing training complexity.

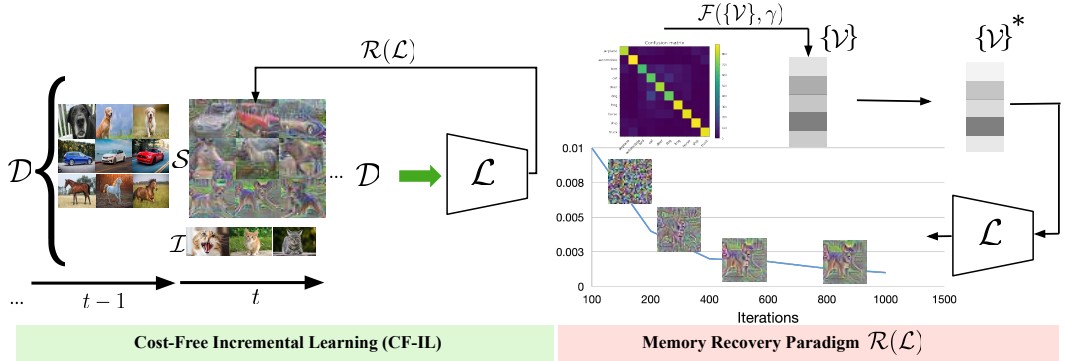

Figure 1: Overview of the proposed approach. When a new incremented task data $I$ emerge, the memory recovery paradigm $\mathcal{R}(\mathcal{L})$ is called to generate transfer set $\mathcal{S}$, including synthesized images for the classes that the learner network $\mathcal{L}$ has learned before. In the proposed recovery engine, we first generate preliminary model outputs $\{\mathcal{V}\}$ for each learned class and then form a refined set $\{\mathcal{V}\}^*$ by applying a constraint $\mathcal{F}(\{\mathcal{V}\}, \gamma)$ using a dynamic recommender vector $\gamma$ to adjust due to underlying distribution. Then, a random noise input is initialized to be optimized conditioned on the refined output $\{\mathcal{V}\}^*$. This procedure is performed for each learned class several times to form the transfer set $\mathcal{S}$. Finally, the learner network is retrained on the combined dataset $\mathcal{D}$, including the transfer set $\mathcal{S}$ and new incremented task data $\mathcal{I}$ using a two-part cost function composing a classification (CE) and knowledge distillation (KD) terms, respectively.

***Regularization-based:*** These works penalize the model for updating important parameters for earlier tasks by adding a regularization term to the loss function. The importance score of each parameter can be estimated through the Fisher Information Matrix (Kirkpatrick et al., 2017), gradient extent (Aljundi et al., 2018), and uncertainty estimation using Bayesian Neural Network (Nguyen et al., 2017). Although the vast amount of works falls into this family, soft penalties might not be sufficient to restrict the optimization process to stay in the feasible region of previous tasks, particularly with long sequences (Farquhar & Gal, 2018), resulting in increased forgetting of earlier tasks.

***Parameter Isolation:*** These methods rely on estimating binary masks that allocate a fixed number of parameters for each task where the network architecture remains static. Such masks can be estimated either by random assignment (Masse et al., 2018), pruning (Mallya & Lazebnik, 2018), or gradient descent (Mallya et al., 2018; Serra et al., 2018). However, this family is limited to the task incremental setting and better agreed for learning a long sequence of tasks when model capacity is not a concern, and optimal performance is the priority.

## 3 PROBLEM SETTING

Without loss of generality, a classical IL scenario can be assumed as a set $T = \{T_1, \ldots, T_N\}$ of $N$ different tasks. Each task $T_i$ takes the form of a classification problem, $T_i = \{\mathbf{x}_j, y_j\}_{j=1}^{i \times l}$, where $y_j \in \{1, \ldots, l\}$. When a new task arrives we ask the network to learn to classify the new images, being simultaneously able to solve the former tasks. Our method only relies on a single DNN network, named the learner network $\mathcal{L}$, in all steps with no external memory and no parallel network. We formalize the setting conditioned on the absence of a task oracle at test time as follows: Task-IL and Class-IL. In the Task-IL setting, the learner network $\mathcal{L}$ is composed of a feature extractor followed by a multiple heads where each head is composed of a linear classifier with a softmax activation which computes task-specific classification probabilities. On the contrary, in the Class-IL setting, the learner network has a single head while is shared over all the tasks that make the Class-IL more challenging. In this paper, we consider both settings and report the results regarding them.

## 4 PROPOSED METHOD

Availability of the actual data samples can enable solving for straightforward retraining, thus learning incrementally can overcome catastrophic forgetting problem. However, preserving earlier data

is not straightforward in real-world scenarios due to computational overhead and privacy concerns. In this paper, the main idea is to recall (instead of preserving) past exemplars, known as network experiences, without using either a parallel network or memory buffer. To this end, we propose CF-IL in which, whenever a new task arises, we trigger a memory recovery paradigm $\mathcal{R}(\mathcal{L})$ to obtain a transfer set $\mathcal{S}$ of past experiences as surrogates for the earlier tasks observations. Then, we form a combined training dataset $\mathcal{D}$ from the recovered transfer set $\mathcal{S}$ and the current incremented class(es) $\mathcal{I}$ data. Thus, the learner network $\mathcal{L}$, which is supposed to learn the task sequences $T$ incrementally, is retrained on $\mathcal{D}$ in each time step. In the suggested Memory Recovery Paradigm (MRP), we first model the learner network output $\mathcal{L}$ regarding the case when a real sample from the past is fed to the model. These outputs are provided by generating initial output vectors $\{\mathcal{V}\}$ and refining them by applying the $\mathcal{F}(\mathcal{V}, \gamma)$ constraint, as a supervision, to perform adjustment due to underlying distribution obtained from a dynamic recommender $\gamma$, resulting in reaching $\{\mathcal{V}\}^*$. Fig. 1 shows a clear overview of the proposed method. In the rest of this section, we first describe the proposed CF-IL and then outline the suggested MRP in detail.

## 4.1 COST-FREE INCREMENTAL LEARNING

In this work, the cost-free incremental learning term expresses that there is no need for a memory buffer to learn incrementally, while the DNN model complexity is fixed. Besides, we synthesize a transfer set $\mathcal{S}$ as past experiences of the network in a cost-free manner. To put this idea into practice, we run the following steps, which are listed in Alg. 1. Whenever CF-IL receives a new incremented task data $\mathcal{I} = \{(x_i, y_i), 1 \leq i \leq L, y_i \in [1, \ldots, l]\}$, where $L$ is the total number of samples of $l$ classes belonging to the new incremented task $T^t$, it initiates the memory recovery procedure $\mathcal{R}(\mathcal{L})$ (Alg. 2) to create the transfer set $\mathcal{S} = \{(\hat{x}_j, \hat{y}_j), 1 \leq j \leq K, \hat{y}_j \in [1, \ldots, k]\}$, where $K$ is the total number of synthesized samples of $k$ classes that have been learned by the learner network $\mathcal{L}$. Next, the samples from the transfer set $\mathcal{S}$ is fed to the learner network, and the resulting network's logits $\mathcal{O} = [o_1(x), \ldots, o_K(x)]$ for all samples are stored. Then, we form a combined dataset $\mathcal{D}$ per each task $T_i, i \in [1, \ldots, N]$ in the task sequence $T$. Finally, the learner network $\mathcal{L}$ parameters are updated to minimize a cost function such a way that each data sample from the new incremented class $\mathcal{I}$ will be classified correctly, as classification loss, and for the samples in $\mathcal{S}$, current network's logits will be reproduced as close as those have been stored in the previous step, as knowledge distillation loss by KD term.

We employ the softmax cross entropy as the classification loss, which is computed using Equ. 1:

$$\ell_{CE}(\mathcal{L}(x; \theta_\mathcal{L}), y) = - \sum_{(x,y) \in \mathcal{I}} \sum_{i=1}^{l} \delta_{y=i} \log [p_i(x)] \tag{1}$$

where $\delta_{y=i}$ is the indicator function and $p_i(x)$ is the output probability (i.e, softmax of logits) of the $i^{th}$ class in $l$ new incremented class(es).

For the distillation purpose, we adopt knowledge distillation from network output (logits), known as dark knowledge Hinton et al. (2015); Wang & Yoon (2020); Buzzega et al. (2020), and our objective is to minimize the Euclidean distance between the stored logits in the transfer set $\mathcal{O} = [o_1(x), \ldots, o_K(x)]$ and those generated by the learner network $\hat{\mathcal{O}} = [\hat{o}_1(x), \ldots, \hat{o}_K(x)]$ as follows:

$$\ell_{KD}(\mathcal{L}(x; \theta_\mathcal{L}), \mathcal{O}) = \sum_{(x,y) \in \mathcal{S}} \sum_{j=1}^{k} \left\| o^j(x) - \hat{o}^j(x) \right\|_2^2 \tag{2}$$

Having a look at both cost functions, we define the total loss function by a linear combination as below:

$$\ell_{Total} = \ell_{CE} + \lambda \ell_{KD} \tag{3}$$

where $\lambda$ is predefined parameter to control the degree of distillation.

## 4.2 MEMORY RECOVERY PARADIGM

The proposed MRP is devised to synthesize the previously learned experiences (knowledge) by $\mathcal{L}$. In particular, we synthesis past samples in a way that the learner network strongly believes them to be actual samples that belong to categories in the underlying data distribution. Therefore, they are

Figure 2: Visualization of the sampled recovered data from the learner network using the proposed memory recovery paradigm, while training on CIFAR10 dataset.

the network acquaintance that might not be natural-looking data. It is worth noting that we only use the learned parameters of the learner model $\mathcal{L}$ since they can be interpreted as the memory of the model in which the essence of training has been saved and encoded.

In the suggested paradigm, we first model the network output space using a two-stepped approach and then generate synthesized samples by back-propagating the modeled output through the network, where each is described in the rest. The whole procedure is shown in Alg. 2.

**Modeling the Network Output Space** In the first phase, we model the output space of the learner network. Suppose $y^{i,j} \sim \mathcal{L}(x, \theta_\mathcal{L})^{(T_i, C_j)}$ is the actual model output when $j^{th}$ class of $i^{th}$ task observation was fed to the network. To reproduce such output in the absence of actual previous one, $(\mathcal{V}^*)^{i,j} \sim y^{i,j}$, we consider a two-stepped approach as is explained bellow.

In (Nayak et al., 2019), it is investigated that sampling the output vector from a Dirichlet distribution in which ingredients are in the range $[0, 1]$ and whose sum is one is a straightforward way for modelling the output space of the learner network. Despite interesting results, this method lacks preserving extra class similarity, hence generating outlier vectors that do not follow the networks underlying distribution. Thus, false memory happens when we query the memory for a specific target. In this case, the result of memory recovery is a mixture of some targets that do not really exist in the past experience, resulting in confusing the learner in the retraining phase. In our novel memory recovery process, we make supervision as the second step, to detect such outliers to boost the retrieved memory in terms of generating a single-target-based transfer set. To do this, we make supervision by applying a constraint on the generated vector $\mathcal{F}(\mathcal{V}, \gamma)$, in which an arbitrary generated vector from the previous step $\mathcal{V}$ is a good candidate only if it has a distance lower than a predefined threshold $\eta$ to a dynamic recommender $\gamma$. This recommendation vector supposed to be declared at a low-cost and represent the class similarity as well. Considering these concerns, we imply a dynamic confusion matrix $CM$ that is constructed incrementally when training each task. An arbitrary confusion matrix is a table that is typically used to describe the performance of a classification model. Intuitively, when a model wrongly classifies some samples to class $c$, it means class $c$ is highly correlated with the target class. On the other side, when a model always correctly classifies class $c$ from the target class, it means class $c$ is highly distinguishable from the target class. This is the exact expected result from a DNN model output. Mathematically, we check if the generated vector $\mathcal{V}^i$ for arbitrary class $i$, has a distance smaller than a predefined threshold $\eta$ to the $i^{th}$ row of the constructed dynamic confusion matrix $CM^i$, so as to obtain the checked output vector $\mathcal{V}^{*i}$.

**Back propagate Through the Network** In the second phase, conditioned on the network's generated outputs $\{\mathcal{V}^*\}$, we synthesize a transfer set $\mathcal{S}$. For an arbitrary generated vector $\mathcal{V}^*$, we start with a random noisy image $x$ sampled from a uniform distribution in the range $[0, 1]$ and update it till the cross-entropy (CE) loss between the generated model output $\mathcal{V}*$ and the model output predicted by the learner network $\mathcal{L}(x, \theta_T, \tau)$ is minimized.

$$\bar{x} = \arg\min \ell_{CE}\left(\mathcal{L}(x, \theta_\mathcal{L}, \tau), \boldsymbol{y}\right) \tag{4}$$

where $\tau$ is the temperature value used in the softmax layer (Hinton et al., 2015) for the distillation purpose. This procedure is renewed for each of the $k$ classes has been learned $K/k$ times, where $K$ is the whole number of samples to be formed the transfer set $\mathcal{S}$.

The proposed paradigm is a remedy for memory bottlenecks so that the model alone can alleviate catastrophic forgetting, resulting in remarkable performance among the data-free works on both task-IL and class-IL, as will be seen in the next section.

**Algorithm 1:** Cost-Free Incremental Learning

---

**Input** : Task sequence set $T$, Transfer set size $K$, Learner model parameters $\theta_{\mathcal{L}}$

**Require:** Confusion Matrix $CM$ from $\mathcal{L}$, Incremented class(es) $\mathcal{I}$

// memory recovery paradigm

$\mathcal{S} \leftarrow \mathcal{R}(\theta_L, K, CM)$

$L$ : number of samples in $\mathcal{I}$

// store network outputs and Ground Truth for $\mathcal{S}$ with pre-update parameters

**for** $(x, y)$ *in* $\mathcal{S}$ **do**
$\quad | \quad \mathcal{O} \leftarrow \mathcal{L}(x; \theta_{\mathcal{L}})$

// form combined training set

$\mathcal{D} \leftarrow \bigcup_{i=1}^{i=L} \{(x^i, y^i, -) : x^i \in \mathcal{I}_t\} \quad \cup$
$\qquad\qquad \bigcup_{j=1}^{j=K} \{(x^j, y^j, \mathcal{O}^j) : x^j \in \mathcal{S}_t\}$

// define a two-parted loss function

$\ell_{Total} =$
$\quad \underset{(x,y,\mathcal{O}) \in \mathcal{D}}{\Sigma} [\Sigma_{i=1}^{N} \ell_{CE}(\mathcal{L}(x^i; \theta_{\mathcal{L}}), y^i)] \quad +$
$\qquad\qquad\qquad \lambda [\Sigma_{j=1}^{K} \ell_{KD}(\mathcal{L}(x^j; \theta_{\mathcal{L}}), \mathcal{O}^j)]$

// optimize the network for the total loss

**for** $(x, y)$ *in* $range(K + L)$ **do**
$\quad$ sample a batch $b_1$ form $\mathcal{D}$, where $(x, y, \mathcal{O}) \in \mathcal{S}$
$\quad$ sample a batch $b_2$ form $\mathcal{D}$, where $(x, y) \in \mathcal{I}$
$\quad$ update $\theta_{\mathcal{L}}$ by taking a SGD step on $b_1 + b_2$
$\quad$ loss $\ell_{Total}(\theta_{\mathcal{L}})$

---

**Algorithm 2:** Mem. Recovery Paradigm

---

**Input** : Learner model parameters $\theta_{\mathcal{L}}$, Transfer set size $K$, Confusion Matrix $CM$

**Output** : Transfer set $\mathcal{S}$

**Require:** Dirichlet distribution parameters $\alpha, \beta$, Thresh. $\eta$, Temperature for distillation $\tau$

$k$ : number of classes from $\mathcal{L}$

$p : K/k$     // Number of samples per class

$\mathcal{S} \leftarrow \{\}$

Constraint $\mathcal{F}(\mathcal{X}, \gamma) \leftarrow (\|\mathcal{X} - \gamma\|_2^2 < \eta)$

**for** $i = 1 : k$ **do**
$\quad \gamma \leftarrow CM^i$
$\quad$ **for** $j = 1 : p$ **do**
$\quad\quad$ **do**
$\quad\quad\quad$ Sample $\mathcal{V}_j^i \leftarrow$
$\quad\quad\quad\quad$ Dir $\left(d, \beta \times \boldsymbol{\alpha}^i\right)$
$\quad\quad\quad \mathcal{V}^{*i}_j \leftarrow \mathcal{V}_j^i$
$\quad\quad$ **while** *Constraint* $\mathcal{F}(\mathcal{V}_j^i, \gamma)$ *passed*
$\quad\quad$ Initialize $x_j^i$ to random noise
$\quad\quad \bar{x}_j^i \leftarrow$ optimize $x_j^i$ with
$\quad\quad\quad\quad\quad \ell_{CE}\left(\mathcal{V}^{*i}_j, \mathcal{L}\left(x_j^i, \theta_{\mathcal{L}}, \tau\right)\right)$
$\quad\quad \mathcal{S} \leftarrow \mathcal{S} \cup \bar{x}_j^i$

---

## 5 EXPERIMENTS

### 5.1 EXPERIMENTAL SETUP

We evaluate our method on two commonly-used datasets for incremental image classification tasks: CIFAR-10 (Krizhevsky et al., 2009) and Tiny-ImageNet (Le & Yang, 2015). To form incremented tasks, we randomly choose 2, 20 classes for each task resulting in a sequence of 5, 20 sequential tasks for CIFAR-10 and Tiny-ImageNet, respectively, which is a typical configuration in literature. Following iCaRL (Rebuffi et al., 2017), a ResNet18 (He et al., 2016) is used in the learner network without pre-training. To perform a fair comparison with other IL methods, we train the networks using the Stochastic Gradient Descent (SGD) optimizer with a learning rate of $0.1$ with other parameters set to their default values. We consider the number of epochs per task concerning the dataset complexity; thus, we set it to 50 for Sequential CIFAR-10 and 100 for Sequential Tiny-ImageNet, respectively. In the training phase, we select a batch of data from the incremented task and a mini-batch of data from the transfer set $\mathcal{S}$, where depending on the hardware restriction, we set both to 32. We show all the classes in the same fixed order across different runs, and results are averaged across ten runs, each involving a different initialization. For visual simplicity, we only report mean values. Following the convention of the ML community, hyperparams are selected by performing a grid-search on a validation set, obtained by sampling $10\%$ of the training set. In the MRP, we consider the temperature value $\tau$ to 20 for the distillation purpose. In the Dirichlet distribution, we set $\beta$ in $[1, 0.1]$ for each dataset, where half the transfer set $\mathcal{S}$ is synthesized by $1$ and the rest with $0.1$ as in (Nayak et al., 2019). The $\eta$ value in the constraint step is empirically valued at $0.7$. To optimize the random noisy image, we employ the Adam optimizer with a learning rate of $0.01$, while the maximum number of iterations is set to $1500$. For all experiments, we consider the average accuracy across tasks as the evaluation criteria.

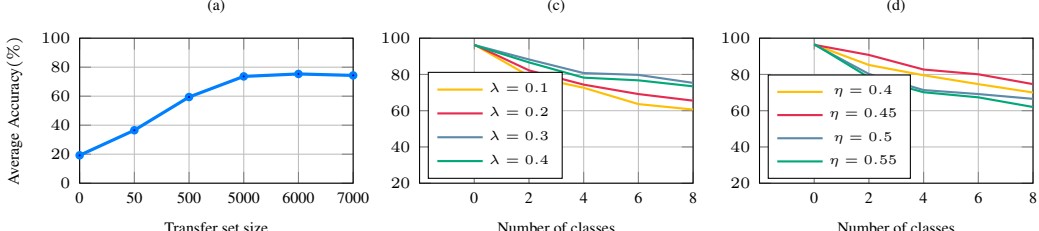

Figure 3: Experimental results on CIFAR-10 dataset. **(a)** classification accuracy curves for CF-IL with various transfer set size. Performance comparison of the CF-IL with various hyperparameters $\lambda$ **(c)** and $\eta$ **(d)**.

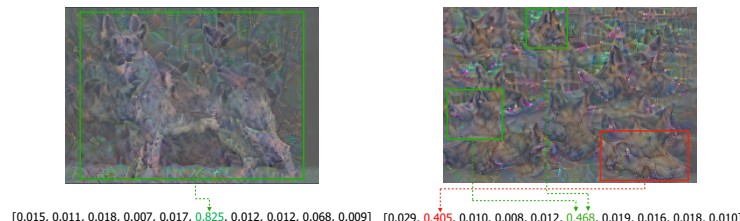

[0.015, 0.011, 0.018, 0.007, 0.017, 0.825, 0.012, 0.012, 0.068, 0.009]  [0.029, 0.405, 0.010, 0.008, 0.012, 0.468, 0.019, 0.016, 0.018, 0.010]

Figure 4: Visualization of happened False Memory when retrieving data from the memory recovery step (Right) and a correct sample which passed the second step in our introduced network output modeling procedure (Left).

## 5.2 ABLATION STUDIES

**Synthesized Transfer Set:** We follow a two-step strategy to generate the network output in MRP. Firstly, we follow the approach adopted in (Nayak et al., 2019) in which a Dirichlet distribution $\text{Dir}(d, \beta \times \boldsymbol{\alpha})$ is utilized to generate the vectors in sotmax space, where we need to set two parameters in this distribution: $\alpha$, and $\beta$ ($d$ is the dimension of the expected vector). $\alpha$ is the concentration parameter to control the probability mass of a sample from the distribution. Decreasing its value caused a vector that is hugely concentrated in only a few ingredients and vice versa. $\alpha$ is interpreted as a class similarity vector obtained from the network's final layer and can be scaled with the $\beta$ parameter. This method encounters false memory since some generated vectors are a mix-up of several targets. To tackle this problem, we suggest applying a constraint to refine these vectors. An examples of false memory and a correct one for the target class dog are shown in Fig. 4. Moreover, we present several samples of retrieved images belonging to several classes, when querying the learner network $\mathcal{L}$ using our novel MRP, as shown in Fig. 2. We can see how the network retains its learned knowledge in its memory which is a specific pattern representing the target classes.

**Hyper-parameters:** $\lambda$, $\eta$ **and** $\mathcal{S}$: $\lambda$ controls how much the transfer set $\mathcal{S}$ should be attended as Eq. 3, and $\eta$ is related to how much the generated model output should be close to the recommendation vector. To select the best one for the two mentioned parameter, we perform grid search, where a brief of examination are reported in Fig. 3 **(c)** and **(d)**, where a take $\lambda$ and $\eta$ as $0.3$ and $0.45$, respectively. We also examine the impact of transfer set size on the performance of the incrementally learning classes. To this end, we set up the proposed CF-IL on CIFAR-10 dataset with different sizes of transfer sets, including $[50, 500, 5000, 6000, 7000]$, wherein in the one before the latest case, the retrieved data from the past is equal to the incremented class. Fig. 3 **(a)** shows the performance. Obviously, increasing the number of synthesized samples in the transfer set has a significant impact on the performance. It is worth noting that an optimal size for the transfer set depends on the task complexity in terms of the number of classes and variations of the actual images. Thus, increasing the set size higher than a reasonable one might increase the risk of overfitting.

**CF-IL in memory-based works:** We conduct a set of experiments on CIFAR-100 by adopting our MRP as a buffer booster which means it can be used as an extension in any buffer-based works. In this case, with having a fixed-sized buffer to store actual data from the past, we add the transfer set

resulting in more balanced data when training on a new arrival task. We also consider two more cases in which no actual data is buffered, and a few samples (10 samples per class) are buffered from the past, specified by buffer size. The evaluation results are reported in the table below, where $N$ means the incremental phases equal the number of tasks. As shown in the table, while having the same IL method, our method is a powerful extension while it has more consistent results when increasing the number of incremental phases means our MRP can synthesise past samples effectively regardless of how many classes have been learned by the learner network.

Table 1: Classification results (average accuracy) on CIFAR-100 dataset for buffer-based methods and ablative models with adopting MRP as a buffer booster.

| Method / N | 5 | 10 | 25 | buffer size |
|---|---|---|---|---|
| iCaRL | 57.12 | 52.66 | 48.22 | 2K |
| iCaRL + mnemonics | 59.88 | 57.53 | 54.30 | 2K |
| LUCIR | 63.17 | 60.14 | 57.54 | 2K |
| LUCIR + mnemonics | 64.95 | 63.25 | 63.70 | 2K |
| iCaRL + MRP | 61.03 | 59.79 | 58.01 | 2K |
| LUCIR + MRP | **65.32** | **65.21** | **65.80** | 2K |
| CF-IL | 62.37 | 62.01 | 61.79 | 0 |
| CF-IL | 63.64 | 63.18 | 63.12 | 1K |

## 5.3 COMPARED METHODS

To have a fair comparison, we first should determine that our work falls into which categories are mentioned in Sec 2. Secondly, discuss two main aspects that should be considered when it comes to comparing methods: Performance, complexity. Obviously, our method is neither model-growing PNN (Rusu et al., 2016) nor parameter-isolation since our learner network has a fixed architecture while the whole parameters are share among all tasks. In terms of replaying past samples, CF-IL may be counted as a member of memory-replay-based group MER (Riemer et al., 2018), iCaRL (Rebuffi et al., 2017), DER (Buzzega et al., 2020), GEM (Lopez-Paz & Ranzato, 2017), a-GEM (Chaudhry et al., 2018), GSS (Aljundi et al., 2019), TPCIL (Tao et al., 2020), Mnemonics (Liu et al., 2020), BiC (Wu et al., 2019) that more recent SOTAs fall into this category. However, keeping past examples in a fixed-size buffer is not applicable when growing the number of tasks and has a memory overhead resulting in increasing complexity of the system. Keeping these issues in mind, we compare our method with prominent works in this category while allocating the same buffer size and omit the buffer for our approach. Our work somehow can be regarded as a data-free work, so it falls into the regularization-based group oEWC (Schwarz et al., 2018), SI (Zenke et al., 2017), ALASSO (Park et al., 2019), UCB (Ebrahimi et al., 2019). Although this category of works performs incremental learning without any equipment, they can't gain outstanding performance, while CF-IL achieves remarkable results. The more comparable group of works is pseudo-memory-rehearsal DGM (Ostapenko et al., 2019), DGR (Shin et al., 2017), MeRGAN (Wu et al., 2018) which try to synthesis past knowledge. Thanks to the generative adversarial networks, this group of papers meet a comparable result against memory-rehearsal works without buffering strategy. However, CF-IL has the same terminology while reducing the complexity by eliminating the need for an auxiliary network along with improving performance.

## 5.4 RESULTS

Table 2 shows the performance of CF-IL in comparison to the other mentioned methods on the two commonly used considered datasets. Our proposed method CF-IL has achieved the SOTA performance in almost all settings. PNN (Rusu et al., 2016), one of the dynamic architecture works, produces the most substantial results in the task-IL setting, specifically 95.13% and 67.84% in CIFAR-10 and tiny-ImageNet, respectively. However, it suffers from an exponential memory increasing issue, making it impossible in class-IL and challenging datasets with more classes. Besides, our method won second place in the Task-IL setting at the cost of keeping the network architecture fixed with margins 2.01% on CIFAR-10 and 0.42% on Tiny-ImageNet than the top rank. Compared with regularization-based methods SI (Zenke et al., 2017), oEWC (Schwarz et al., 2018), and ALASSO (Park et al., 2019) the gap does not seem to be bridged, indicating that the old parameter set's regu-

Table 2: Classification results (average accuracy) for IL benchmarks, averaged across 10 runs. "-" denotes experiments we were unable to run, because of compatibility issues. Best performance is marked in bold in each section. We use public code (Buzzega et al., 2020) to reproduce results. Sections show joint training, model growing, parameter isolation, memory replay, regularization, and pseudo memory replay from the top.

| Method | CIFAR-10 | | Tiny-ImageNet | |
|---|---|---|---|---|
| | Class-IL | Task-IL | Class-IL | Task-IL |
| JOINT | 92.02 | 98.29 | 59.45 | 82.07 |
| PNN | - | **95.131** | - | **67.84** |
| HAL | 41.79 | 84.54 | - | - |
| MER | 57.74 | 93.61 | 9.99 | 48.64 |
| GEM | 26.20 | 92.16 | - | - |
| a-GEM | 22.67 | 89.48 | 8.06 | 25.33 |
| iCaRL | 47.55 | 88.22 | 9.38 | 31.55 |
| DER | **70.51** | 93.40 | 17.75 | **51.78** |
| BiC | 49.18 | 89.91 | 11.09 | 32.94 |
| WA | 53.76 | 94.21 | 15.42 | 37.12 |
| Mnemonics | 57.26 | 95.91 | 23.47 | 42.36 |
| TPCIL | 58.72 | **96.67** | **25.03** | 48.41 |
| oEWC | 19.49 | 68.29 | 7.58 | 19.20 |
| SI | 19.48 | 68.05 | 6.58 | 36.32 |
| ALA | 25.19 | 73.79 | 17.02 | 48.07 |
| UCB | **56.23** | **78.56** | **23.43** | **49.01** |
| DGM | 71.94 | 89.91 | **28.45** | 61.32 |
| DGR | 49.69 | 79.86 | 17.38 | 38.41 |
| MeRGAN | 66.76 | 84.76 | 23.86 | 58.32 |
| **CF-IL (Ours)** | **75.34** | **93.12** | **33.06** | **67.42** |

larization is not sufficient to mitigate forgetting since the wights' importance, calculated in an earlier task, might be unreliable in later ones. Even though CF-IL works without allocating memory buffer, our method can achieve excellent results in both class-IL and task-IL on both datasets. In particular, our proposed method obtains 75.43%, 93.12% in the class-IL, Task-IL settings on the CIFAR-10, and 733.06%, 67.42% in the class-IL, Task-IL settings on the Tiny-ImageNet, respectively, making the relative improvement over the recent SOTA TPCIL (Tao et al., 2020) by 16.62%, 8.03% in the class-IL settings on the CIFAR-10 and Tiny-ImageNet, and 19.01% in the task-IL in the more challenging Tiny-ImageNet respectively. We further investigate the impact of buffering samples against synthesizing samples in the ablation study. Compared to pseudo-rehearsal-based methods, which have recently gained much attention due to the power of simulating past experiences rather than storing them, the proposed method shows a significant outperformance in both accuracy and processing time since there is no auxiliary network needed. In particular, our method achieves a relative improvement of 3.40%,3.21%,4.61%,6.1% over DGM (Ostapenko et al., 2019).

## 6 CONCLUSION

We have proposed a novel strategy for IL to address the memory issue, which is crucial when the number of classes becomes large. In particular, we perform IL in both class-IL and task-IL settings in a cost-free manner. This strategy is implemented through a memory recovery paradigm with no additional equipment. It only relies on the single DNN, known as the learner, to retrieve the network's past knowledge as a transfer set to look back on learned experiences. Our method has outstanding results, compared with recent prominent works.

## 7 ACKNOWLEDGEMENT

The authors would like to thank Part Research Center (Partdp.ai) for contributing to the hardware infrastructure we used for our experiments.

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
