# OpenReview forum: "Looking Back on Learned Experiences  For Class/task Incremental Learning"
_ICLR.cc/2022/Conference — ICLR 2022 Spotlight_

### Official Review · Reviewer_ruLe · 2021-11-04

**Correctness:** 3
**Technical Novelty And Significance:** 2
**Empirical Novelty And Significance:** 2
**Recommendation:** 6
**Confidence:** 3

**Main Review:**

- The proposed method takes an advantage of generative model-based approach without an explicit generative model.

- Experimental results looks strong.

- Discussion on memory replay methods is incomplete or misleading. Recent works (roughly, those published in 2019 or later) proposed to solve the bias problem, while the statement in this paper sounds the solution is not investigated. There are some early works addressing the bias problem but missing in this work: [Castro et al.], [Javed et al.], [Lee et al.].

- The proposed method is not really zero-shot "learning"; In zero-shot learning, your model has never seen zero-shot classes but inferred them from auxiliary information.  It seems they argue so because their method has no extra memory or generative model, but I believe zero-shot learning is not the term for this case, thus misleading. The experimenal setting of LwF (though later works added extra memory to keep previously seen data in their experiments for fair comparison) or the usage of unlabeled data in [Lee et al.] (again, though they also added extra memory for previously seen data for fair comparison) also do not require an extra memory or generative model, but they have never claimed that their method is zero-shot (incremental) learning. I found that the zero-shot knowledge distillation work is cited, but their work is about zero-shot "knowledge distillation," not zero-shot learning. I found that this paper did not cite any zero-shot learning works; I recommend [Xian et al.] to read.

- More ablation study is required to convince the experimenal results: for example, the choice of the prior distribution for the recommender vector (even one-hot + Gaussian noise + normalization could be a simple baseline), the learning schedule for the memory recovery step (by the way, you should also report how long this step takes, otherwise your method has an unfair advantage on the training time than other compared methods), the size of the synthesized dataset, and so on.

- In Table 2, I recommend to put the name of each types, either in the caption or by adding a column; otherwise it is hard to figure out how they are grouped, unless you fully memorize the related work section and the abbreviation of all prior works.

[Castro et al.] End-to-end incremental learning. In ECCV, 2018.\
[Javed et al.] Revisiting Distillation and Incremental Classifier Learning. In ACCV, 2018.\
[Lee et al.] Overcoming Catastrophic Forgetting With Unlabeled Data in the Wild. In ICCV, 2019.\
[Xian et al.] Zero-Shot Learning -- A Comprehensive Evaluation of the Good, the Bad and the Ugly. TPAMI, 2018.

## Post rebuttal

As the authors addressed most of my concerns, I change my recommendation to 6: marginally above the acceptance threshold.


**Summary Of The Paper:**

This paper proposes to generate pseudo data using the gradient-based method for class- and task-incremental learning. Experimental results show the effectiveness of the proposed method.

**Summary Of The Review:**

The method is simple yet effective, which is good. However, "zero-shot" sounds misleading, and ablation study is not enough.

---

> ### Author Response · Authors · 2021-11-17
> **Discussion on memory replay,  Zero-short, more ablation study, computational cost**
>
> Q1:Discussion on memory replay methods is incomplete or misleading. Recent works (roughly, those published in 2019 or later) proposed to solve the bias problem, while the statement in this paper sounds the solution is not investigated. There are some early works addressing the bias problem but missing in this work: [Castro et al.], [Javed et al.], [Lee et al.].
> R1:Thank you for taking the time to review the literature related to our work. We need to revise to clarify the assertion. It is better to say that the use of buffering strategies will lead to bias problems. Some recent work uses distillation loss or unlabeled data to solve this problem, thereby improving learning performance.
>
> Q2:The proposed method is not really zero-shot "learning"; In zero-shot learning, your model has never seen zero-shot classes but inferred them from auxiliary information. It seems they argue so because their method has no extra memory or generative model, but I believe zero-shot learning is not the term for this case, thus misleading. The experimenal setting of LwF (though later works added extra memory to keep previously seen data in their experiments for fair comparison) or the usage of unlabeled data in [Lee et al.] (again, though they also added extra memory for previously seen data for fair comparison) also do not require an extra memory or generative model, but they have never claimed that their method is zero-shot (incremental) learning. I found that the zero-shot knowledge distillation work is cited, but their work is about zero-shot "knowledge distillation," not zero-shot learning. I found that this paper did not cite any zero-shot learning works; I recommend [Xian et al.] to read.
>
> R2:Thanks for calling it to our attention. We need to make this change so as not to cause further misunderstandings. As you pointed out, we called our method "Zero_Shot changed to Data-Free" since we are able to replay past samples with no need for any memory for buffering data or auxiliary network to generate data. Therefore, our method obviously does not belong to the field of zero-sample learning, as reviewer #1 and reviewer #2 pointed out. The naming is changed in the new rebuttal revised paper.
>
> Q3:More ablation study is required to convince the experimenal results: for example, the choice of the prior distribution for the recommender vector (even one-hot + Gaussian noise + normalization could be a simple baseline), the learning schedule for the memory
> recovery step (by the way, you should also report how long this step takes, otherwise your method has an unfair advantage on the training time than other compared methods), the size of the synthesized dataset, and so on.
>
> R3:We appreciate your suggestion. It is worth noting to make a quick review of the employed memory recovery paradigm. Firstly, we try to model the network output space using Dirichlet distribution to simulate a vector with a distribution like when actual past data was fed to the learner network before learning a new task. Secondly, we refine our generated vector based on a recommender vector to eliminate those vectors that cause false memory problems. Finally, in order to have a low-cost strategy, we use each row of an incremented confusion matrix for each class as the recommender vector for that class and keep those generated vectors if they have a distance smaller than a predefined threshold. Then, the vectors generated by the refinement are backpropagated through the network to synthesize images that the learner network believes those vectors cause when feeding such images. If we truly understand, you suggest having some other strategy for generating output vectors compared to using Dirichlet distribution. Actually, we have no particular baseline in previous works to compare with them. Moreover, the intuition behind generating network output is simulating model-specific output vector regarding class similarities which the model has learned. So, a baseline like "one-hot + Gaussian noise + normalization" does not make sense to have any intuition for this purpose.
>
> Also, your carefully thought-out comment leads to the discussion of computational costs in the paper. In our memory recovery paradigm (MRP), all layers of the learner network are frozen in the propagation step, and only the noisy initialized input will be optimized to become a synthetic image conditioned on the output vector generated in the modeling output network phase. Therefore, in the case of CIFAR-10 incremental training, in the MRP step, each round of optimization will optimize 32 \* 32 \* 3 parameters, and each image needs 1500 rounds of optimization, resulting in ~4M parameters update, which is less than one epoch of training learner network (Resnet-18 with 11.19M parameters) with batch size 1. So, synthesising past samples does not have much impact on training time, and at the same time, brings significant gains.

---

> > ### Author Response · Authors · 2021-11-17
> > **Continue....**
> >
> > Q4:In Table 2, I recommend to put the name of each types, either in the caption or by adding a column; otherwise it is hard to figure out how they are grouped, unless you fully memorize the related work section and the abbreviation of all prior works.
> >
> > R4:Thanks for the suggestion. Table 1 is reported with more clarification in the new rebuttal revised paper

---

> > > ### Comment · Reviewer_ruLe · 2021-11-29
> > > **Additional comments**
> > >
> > > Thanks for your response.
> > >
> > > R2:Thanks for calling it to our attention. We need to make this change so as not to cause further misunderstandings. As you pointed out, we called our method "Zero_Shot changed to Data-Free" since we are able to replay past samples with no need for any memory for buffering data or auxiliary network to generate data.
> > > * Data-free sounds better, but it would not be very distinguishable with generative replay methods. I have no good suggestion, but you may want to emphasize that your method does not need an auxiliary generative model.
> > >
> > > R3: Actually, we have no particular baseline in previous works to compare with them. Moreover, the intuition behind generating network output is simulating model-specific output vector regarding class similarities which the model has learned. So, a baseline like "one-hot + Gaussian noise + normalization" does not make sense to have any intuition for this purpose.
> > > * I would say "to model the network output space using Dirichlet distribution to simulate a vector with a distribution like when actual past data was fed to the learner network before learning a new task" is as questionable as the baseline I provided as an example. Both ignores the correlation among classes, which might be hard to model; the Dirichlet distribution would be a baseline that ML/Stat people can think of while I suggested is a baseline that CS people can think of. My suggestion here is either you justify your choice with either an empirical study or a theoretical proof.

---

### Official Review · Reviewer_WhxT · 2021-11-05

**Correctness:** 3
**Technical Novelty And Significance:** 2
**Empirical Novelty And Significance:** 3
**Recommendation:** 8
**Confidence:** 4

**Main Review:**

### Strengths

&nbsp;

- This paper is well-organized and easy to follow.
- The proposed method is technically sound.
- Extensive experiment results are provided.

&nbsp;

### Weaknesses

&nbsp;

- ***Replaying synthesised samples for old classes is not a novel idea.*** It has been proposed in [A]. In my view, the proposed memory recovery paradigm is very similar to the “data-free generative replay” strategy in [A].

&nbsp;

- ***The authors only provide experiment results on small-scale datasets, e.g., CIFAR-10 and Tiny-ImageNet.*** However, we usually use larger datasets to evaluate incremental learning methods. For example, iCaRL (Rebuffi et al., 2017), BiC (Wu et al., 2019), Mnemonics (Liu et al., 2020) run experiments on CIFAR-100 and ImageNet-1k in their original papers. I don’t see why the authors don’t follow the previous work and choose small-scale datasets.

&nbsp;

- ***Updating synthesised samples requires additional computational costs.*** The authors should compare the training time of their method with related work. If their method requires much additional training time, it is not practical.

&nbsp;

- ***The authors reduced the margin, which might practically violate the length limit.*** For example, the margin in Section 6 is significantly different from the ICLR template.

&nbsp;

- ***There are some minor formatting issues in this paper,*** for example, in the last paragraph on Page 4. The authors should carefully check the paper and make sure there is no such issue in the revision.

&nbsp;

- ***It would be better to provide a reproducibility statement in the paper.*** The authors provide a GitHub link, but I cannot see the code using that link.

&nbsp;

### Reference
*[A] Choi, Yoojin, Mostafa El-Khamy, and Jungwon Lee. "Dual-Teacher Class-Incremental Learning With Data-Free Generative Replay." CVPR 2021 Workshops.*

**Summary Of The Paper:**

In order to achieve exemplar-free incremental learning, the authors introduce a memory recovery paradigm to synthesize past exemplars. They also propose to evaluate incremental learning based on the zero-shot learning setting. Extensive experiment results are provided to show the effectiveness of the proposed method.

**Summary Of The Review:**

Overall, I think this paper proposes an interesting method and provides extensive experimental results. So I tend to accept this paper. However, similar ideas have appeared in the existing work, such as [A]. Many details of this paper are not polished. So the overall score is “borderline accept.”

[A] Choi, Yoojin, Mostafa El-Khamy, and Jungwon Lee. "Dual-Teacher Class-Incremental Learning With Data-Free Generative Replay." CVPR 2021 Workshops.

&nbsp;

### === Post-rebuttal Comments===
The authors addressed most of my concerns in the feedback. They also provided the additional experimental results I asked for. I recommend acceptance.

The authors should include the additional results and corresponding analyses in the next revision.

---

> ### Author Response · Authors · 2021-11-17
> **Novelt , More Experiments, computational costs, Template, GitHub source code**
>
> Q1:Replaying synthesised samples for old classes is not a novel idea. It has been proposed in [A]. In my view, the proposed memory recovery paradigm is very similar to the “data-free generative replay” strategy in [A].
>
> R1:As we mentioned in the related work section, some pseudo-replay-based methods adopt an auxiliary network (like GANs) to generate pseudo samples and replay them as past samples when learning a new task. While synthesising past samples is a common approach to this category of works, the remarkable difference is that we synthesize past samples with no auxiliary equipment, resulting in more efficient incrementally learning. The interesting paper you mentioned falls into pseudo-replay-based methods as the claim that "We propose data-free generative replay (DF-GR) to mitigate catastrophic forgetting in CIL by using synthetic samples from a generative model". So, we can put their result in Table 1, in the second section from the bottom of the table, which is a recent prominent work in this category. Plus, we have strong evidence that we published our work earlier than this paper and do not present it now because it violates the law of blindness.
>
> Q2:The authors only provide experiment results on small-scale datasets, e.g., CIFAR-10 and Tiny-ImageNet. However, we usually use larger datasets to evaluate incremental learning methods. For example, iCaRL (Rebuffi et al., 2017), BiC (Wu et al., 2019), Mnemonics (Liu et al., 2020) run experiments on CIFAR-100 and ImageNet-1k in their original papers. I don’t see why the authors don’t follow the previous work and choose small-scale datasets.
>
> R2:In order to make a comprehensive comparison, we decided to put our methods against outstanding works of each category of approaches (Regularization, memory-replay, etc.). In the meantime, we follow DER [1] to reproduce the results of other works, which report all on the two datasets we also reported in the paper. However, we plan to give it a try and upload the result on CIFAR-100 before 20 Nov.
>
> [1] : Buzzega, Pietro, et al. "Dark experience for general continual learning: a strong, simple baseline."  NeurIPS (2020).
>
> Q3:Updating synthesised samples requires additional computational costs. The authors should compare the training time of their method with related work. If their method requires much additional training time, it is not practical.
>
> R3:Your carefully thought-out comment lead to further discussion of computational costs in the paper. In our memory recovery paradigm (MRP), all layers of the learner network are frozen in the propagation step, and only the noisy initialized input will be optimized to become a synthetic image conditioned on the output vector generated in the modelling output network phase. Therefore, in the case of CIFAR-10 incremental training, in the MRP step, each round of optimization will optimize 32 \* 32 \* 3 parameters, and each image needs 1500 rounds of optimization, resulting in ~4M parameters update, which is less than one epoch of training learner network (Resnet-18 with 11.19M parameters) with batch size 1. So, synthesising past samples does not have much impact on training time, and at the same time, brings significant gains.
>
> Q4:The authors reduced the margin, which might practically violate the length limit. For example, the margin in Section 6 is significantly different from the ICLR template.
>
> R4:As you pointed out, we tried to do more experiments as much as we can to show the effectiveness of our method. However, the updated version came back to the original template regarding margins.
>
> Q5:There are some minor formatting issues in this paper, for example, in the last paragraph on Page 4. The authors should carefully check the paper and make sure there is no such issue in the revision.
>
> R5:Thank you for taking the time to review our paper carefully. This is done in the new rebuttal revised paper.
>
> Q6:It would be better to provide a reproducibility statement in the paper. The authors provide a GitHub link, but I cannot see the code using that link.
>
> R6:We add extra implementation details in Sec 5.1 to make it more reproducible. Also, we did not publish the code for not violating the blindness law. It links to the code would work right after publishing the paper, certainly.

---

> > ### Comment · Reviewer_WhxT · 2021-11-18
> > **Thanks for the authors' feedback**
> >
> > Thanks for the authors' feedback.
> >
> > In general, I am satisfied with the answers. And I am looking forward to the revision.
> >
> > The authors claimed that "they have strong evidence that they published their work earlier than [A]." However, as I am not possible to verify this claim, I still ask the authors to discuss [A] as a closely-related existing work in the revision. If the authors have any concerns, I think they should discuss them with the AC or the program chairs.

---

> > > ### Author Response · Authors · 2021-11-22
> > > **New rebuttal revised paper,  More Experiments**
> > >
> > > Thank you for taking the time to review our paper carefully. We discussed Method [A] in the paper (Section 2). In a short time, we experiment on CIFAR-100, and the results are reported here. Also, it can be placed on paper after acceptance. As shown in the table, our method achieves state-of-the-art in the class incremental learning and comparable results on the task incremental learning. It is worth noting that results are highly dependent on hyperparameters, where it would be better if we had more time to do more experiments.
> > >
> > > * CPT stands for the number of Classes Per Task
> > > | | Class-IL | | Task-IL | |
> > > |:-----------:|:--------:|:-----:|:-------:|:-----:|
> > > | CPT* | 5 | 10 | 5 | 10 |
> > > | oEWC | 15.32 | 28.29 | 19.76 | 29.65 |
> > > | iCaRL | 22.32 | 26.12 | 48.46 | 59.12 |
> > > | Mnemonics | 35.41 | 44.93 | 62.73 | 75.20 |
> > > | a-GEM | 23.54 | 37.62 | 55.23 | 62.21 |
> > > | BIC | 28.12 | 38.36 | 61.76 | 68.47 |
> > > | DER | 39.71 | 49.87 | 62.51 | 74.01 |
> > > | DF-IL(Ours) | 42.39 | 51.87 | 61.01 | 73.31 |
> > >
> > > [A] Choi, Yoojin, Mostafa El-Khamy, and Jungwon Lee. "Dual-Teacher Class-Incremental Learning With Data-Free Generative Replay." CVPR 2021 Workshops.

---

> > > > ### Comment · Reviewer_WhxT · 2021-11-22
> > > > **A question on the results.**
> > > >
> > > > In the table, the results of iCaRL are 22.32 and 26.12, respectively.
> > > >
> > > > However, in the original paper, the results should be higher than 40% (you may see the results in Figure 2a in [iCaRL's paper](https://arxiv.org/abs/1611.07725)).
> > > >
> > > > Could you please explain why your results are so much lower compared to the original paper?

---

> > > > > ### Author Response · Authors · 2021-11-22
> > > > > **Discussion about the results**
> > > > >
> > > > > Thank you for your attention to our results and prompt feedback. In memory replay-based methods, the buffer size has a direct effect on the result. Obviously, increasing buffer size improves the incremental learning performance, especially in the case of having more classes in each task. Also, as we mentioned in the paper, having the buffer is a crucial concern where more recent methods have investigated using a smaller buffer. Particularly, iCaRl (as a preliminary work) set the buffer size to 2000 (Sec 4: iCaRL implementation). Still, recent works use a smaller buffer (usually 200 or 500) to further show the effectiveness of their model, which is capable of training incrementally with no significant performance degradation. In addition, decreasing buffer size caused more research areas in the IL: memory-replay-based field, like selecting more representative samples in buffer using corsets, keeping those samples to preserve data topologies, and etc., which is not a concern of this paper. However, from our point of view, IL needs a complete survey to experiment on various configurations to avoid further misleading. All in all, we use the same configuration for all reporting methods to have a fair comparison. Also, we can report our result with more configuration in the next round of revision. Furthermore, this paper [B] investigates the effect of buffer size with more experiments. Moreover, paper [C]: Table 1 (one example from the recent works) has the same configuration and result on iCaRL as ours, which verifies the correctness of our experiments.
> > > > >
> > > > > [B] Buzzega, P., Boschini, M., Porrello, A., Abati, D. and Calderara, S., 2020. "Dark experience for general continual learning: a strong, simple baseline." NeurIPS 2020.
> > > > >
> > > > > [C] Joseph, K. J., and Vineeth N. Balasubramanian. "Meta-consolidation for continual learning." NeurIPS 2020.

---

> > > > > > ### Comment · Reviewer_WhxT · 2021-11-22
> > > > > > **Thanks for the feedback**
> > > > > >
> > > > > > Thanks for the feedback from the authors.
> > > > > >
> > > > > > I think setting the buffer size as "2000 samples" or "20 samples per class" is the most common setting. It is applied as the default setting in many methods you compared in the table, e.g., iCaRL, Mnemonics, BIC, and DER.
> > > > > >
> > > > > > It is okay to explore the performance when using a smaller buffer size. However, you need to provide the results using the standard setting as well. Otherwise, you cannot claim your method is state-of-the-art.
> > > > > >
> > > > > > I think you are still allowed to add comments before 29th Nov. In my experience, it won't take a long time to run experiments on CIFAR-100. If you can provide the results using the standard setting (buffer size: 20 samples per class) on CIFAR-100, and show your method achieves state-of-the-art performance, I am happy to upgrade the rating.

---

> > > > > > > ### Author Response · Authors · 2021-11-22
> > > > > > > **More discussion about the results**
> > > > > > >
> > > > > > > Thank you for your following our paper. There are a few things worth to be mentioned. We follow the DER (buffer size (bf) is in [200, 500, 5120]) to evaluate our method on Tiny-ImageNet (100'000 samples with 200 classes which is more complex than CIFAR-100 with 60'000 samples with 100 classes) with various configurations. The results on different bf show that our method marginally outstands when bf is in [200, 500] and comparable when bf is 5120 (much more than 2000). In detail, we meet performance with a difference of around ~10% compared to Mnemonics, which is the current state of the art and marginally better than most methods. But, we still meet a better result with a margin of ~5% compared to Mnemonics when we experiment on using a buffer with 500 samples plus with our synthesised transfer set (our DA-IL version mentioned in the paper). However, our method with no buffering strategy has a power of comparability with extremely less equipment (no buffer, no auxiliary network, no more training from scratch) which is the most valuable in the real world when data privacy is a concern.
> > > > > > >
> > > > > > > To sum up, if performance is the main concern, we can propose our method as a buffer booster since by decreasing the buffer size which Mnemonics needs by 0.75% (500 instead of 2000), we achieve better performance with a margin of ~5%. And, if the memory cost is the main concern, we introduce our method as data-free IL since, with much less equipment, we are better than Mnemonics with a significant margin (25.06 in Table 1 in our paper) when its buffer size is 500.
> > > > > > >
> > > > > > > If still more experiment is needed, we can do them in a few days.

---

> > > > > > > > ### Comment · Reviewer_WhxT · 2021-11-23
> > > > > > > > **Thanks for the feedback**
> > > > > > > >
> > > > > > > > Thanks for the further feedback.
> > > > > > > >
> > > > > > > > My primary concern is that the current results are not directly comparable with the numbers provided in the original papers of the related work. As I mentioned, many related papers (e.g., iCaRL, Mnemonics, [PODNet](https://arxiv.org/abs/2004.13513), [LUCIR](https://openaccess.thecvf.com/content_CVPR_2019/papers/Hou_Learning_a_Unified_Classifier_Incrementally_via_Rebalancing_CVPR_2019_paper.pdf), etc.) provide the results on CIFAR-100 and ImageNet-1k and save 20 samples per class.
> > > > > > > >
> > > > > > > > Without the additional results, I am still positive about this paper. But I think I will tend to keep my initial rating.
> > > > > > > > If you can provide the results, I think I can upgrade my rating to eight.

---

> > > > > > > > > ### Author Response · Authors · 2021-11-23
> > > > > > > > > **Thank you for your feadback**
> > > > > > > > >
> > > > > > > > >  We will try to report the results on mentioned datasets by saving 20 samples, and conduct a comprehensive comparison with the state of the art methods by the 26 Nov. Is this convincing  for you? We will put the results here as a comment.
> > > > > > > > >
> > > > > > > > > Also, please consider that this paper simply reduce the cost of "reply based methods"  which is very important.

---

> > > > > > > > > > ### Comment · Reviewer_WhxT · 2021-11-23
> > > > > > > > > > **Thanks for the feedback**
> > > > > > > > > >
> > > > > > > > > > I understand your point that reducing the cost of "reply-based methods" is very important.
> > > > > > > > > >
> > > > > > > > > > I am looking forward to your results and suggest you put the results in both settings in the next revision.

---

> > > > > > > > > > > ### Author Response · Authors · 2021-11-27
> > > > > > > > > > > **More experiments**
> > > > > > > > > > >
> > > > > > > > > > > 	We conduct a set of new experiments on CIFAR-100 in which we follow the mnemonics configuration (experimental settings, network architecture) and use their official code to evaluate and reproduce results. The evaluation results are reported in the table below, where N means the incremental phases (number of tasks). We report the DF-IL as our Data-Free Incremental Learning version in which no actual data is buffered from the past. Also, we present the DA-IL as our Data-Allowed Incremental Learning version in which a few samples (10 samples per class) are buffered from the past to be gathered with our synthesised transfer set in the incremental learning process. Moreover, we experimented with iCaRL and LUCIR  while boosting their buffer with our transfer set in the table as iCaRL+ (DA-IL) and LUCIR + (DA+IL), respectively. As shown in the table, while having the same IL method, our method is a powerful extension when compared to mnemonics.  Furthermore, it is worth noting that our method has more consistent results when increasing the number of incremental phases means our method can synthesise past samples effectively regardless of how many classes have been learned by the learner network.
> > > > > > > > > > >
> > > > > > > > > > > | N                 |   5   |   10  |   25  | buffer size |
> > > > > > > > > > > |-------------------|:-----:|:-----:|:-----:|:-----------:|
> > > > > > > > > > > | LWF               | 49.59 | 46.98 | 45.51 |      2K     |
> > > > > > > > > > > | LWF + mnemonics   | 54.43 | 52.67 | 51.75 |      2K     |
> > > > > > > > > > > | iCaRL             | 57.12 | 52.66 | 48.22 |      2K     |
> > > > > > > > > > > | iCaRL + mnemonics | 59.88 | 57.53 | 54.30 |      2K     |
> > > > > > > > > > > | BiC               | 59.36 | 54.20 | 50.00 |      2K     |
> > > > > > > > > > > | BiC + mnemonics   | 60.67 | 58.11 | 55.51 |      2K     |
> > > > > > > > > > > | LUCIR             | 63.17 | 60.14 | 57.54 |      2K     |
> > > > > > > > > > > | LUCIR + mnemonics | 64.95 | 63.25 | 63.70 |      2K     |
> > > > > > > > > > > | iCaRL + (DA+IL)   | 61.03 | 59.79 | 58.01 |      2K     |
> > > > > > > > > > > | LUCIR + (DA+IL)   | 65.32 | 65.21 | 65.8  |      2K     |
> > > > > > > > > > > | DF-IL (ours)      | 62.37 | 62.01 | 61.79 |      0      |
> > > > > > > > > > > | DA-IL (ours)      | 63.64 | 63.18 | 63.12 |      1K     |

---

> > > > > > > > > > > > ### Comment · Reviewer_WhxT · 2021-11-27
> > > > > > > > > > > > **Thanks for the feedback.**
> > > > > > > > > > > >
> > > > > > > > > > > > Thanks for the additional results provided by the reviewers.
> > > > > > > > > > > >
> > > > > > > > > > > > I am satisfied with the authors' feedback, and I don't have more questions.

---

### Official Review · Reviewer_Q41Z · 2021-11-06

**Correctness:** 3
**Technical Novelty And Significance:** 3
**Empirical Novelty And Significance:** 3
**Recommendation:** 6
**Confidence:** 4

**Main Review:**

Strengths
- It is a cool idea to build a continual learning method that doesn't grow the model size over time or rely on external memory.  The proposed method distills information from the past networks to form a synthetic set of past concepts. Although the example synthesis method is largely based on the Zero-Shot Knowledge Distillation (Nayak et al. 2019), embedding this technique into a continual learning framework is very reasonable and novel.
- The paper is well written and the structure is clear.  The clarity of `Modeling the Network Output Space` can be improved though as it's unclear how exactly to synthesize images from the past weights.

Cons
- "our method needs no fixed-size memory" - This might be inaccurate as the model still needs to store the network from previous tasks (or the teacher network). It doesn't need external memory to store past examples or meta data.
- Missing ablation study on example synthesis.  Is image synthesis necessary? Can you only synthesize features?  How fast is the synthesis procedure? It seems the computation is still an issue since this approach requires synthesizing images first and then training on the new tasks.  It might make sense to compare with existing approaches on computation, training time, etc.
- Missing ablation study on the hyperparameters. How are the temperature, number of synthetic samples,  lambda is chosen?
- Missing baselines in FS-IL and ZS-IL.  Table 1 reports the results in FS and ZS settings but didn't refer to previous methods for comparison. This Github link maintains recent works in this line. https://github.com/zhoudw-zdw/Awesome-Few-Shot-Class-Incremental-Learning
- Missing comparison with the synthesis approaches with metadata.  How well does the method perform in terms of image realism? For example, ACE uses the mean and std of the past data to synthesize images. (https://openaccess.thecvf.com/content_ICCV_2019/html/Wu_ACE_Adapting_to_Changing_Environments_for_Semantic_Segmentation_ICCV_2019_paper.html)
- It might be better to try the approach on more benchmarks like CIFAR-100,  CUB, iNaturalist, etc.







**Summary Of The Paper:**

This paper proposes a zero-shot incremental learning approach that doesn't store past examples or metadata for experience reply. It synthesizes past experiences from the network parameters through a memory recovery paradigm.  The proposed method doesn't rely on external memory or parallel networks and achieves very competitive results on Task-IL and Class-IL tasks.

**Summary Of The Review:**

This paper provides a very interesting and novel idea for continual learning. However, the empirical evaluation and ablation study seem to be weak.

=== Post-rebuttal Comments===
The authors addressed most of my concerns in the feedback.  I kept my score and leaned towards acceptance.

---

> ### Author Response · Authors · 2021-11-17
> **Needs to fixed size memory + Ablation study+ Synthesis features +other datasets**
>
> Q1:"our method needs no fixed-size memory" - This might be inaccurate as the model still needs to store the network from previous tasks (or the teacher network). It doesn't need external memory to store past examples or meta data.
>
> R1:Thank you for calling our attention to the clarification in the claim. To put it better, our method only requires a fixed size of memory to maintain the learner network. Thus, as a significant benefit, neither a memory buffer to keep past samples nor extra memory allocation for network growth is needed, which are the concerns when it comes to dealing with memory footprint in incremental learning.
>
> Q2: Missing ablation study on example synthesis. Is image synthesis necessary? Can you only synthesize features? How fast is the synthesis procedure? It seems the computation is still an issue since this approach requires synthesizing images first and then training on the new tasks. It might make sense to compare with existing approaches on computation, training time, etc.
>
> It seems that you are fully following our approach. The main idea is to ask the network to remember what it has learned in either a feature vector or a synthesized image where we tried both, and there are some notes worth to be mentioned. In our memory recovery paradigm (MRP), all layers of the learner network are frozen in the propagation step, and only the noisy initialized input will be optimized to become a synthetic image conditioned on the output vector generated in the modeling output network phase. Therefore, in the case of CIFAR-10 incremental training, in the MRP step, each round of optimization will optimize 32 \* 32 \* 3 parameters, and each image needs 1500 rounds of optimization, resulting in ~4M parameters update, which is less than one epoch of training learner network (Resnet-18 with 11.19M parameters) with batch size 1. So, synthesising past samples does not have much impact on training time, and at the same time, brings significant gains. Considering the frozen network in MRP, there is no significant difference in computational overhead in propagating the gradient to the input image or the intermediate layer of the network. While the synthetic input image can bring visual intuition to discover what the experience of the network is and help to figure out the recovering process to avoid false memory problems, which is explained in Sec 5.2 in the paper.
>
> Q3:Missing ablation study on the hyperparameters. How are the temperature, number of synthetic samples, lambda is chosen?
> R3:We conducted some experiments on the number of synthetic samples and lambdas reported in Figure 3 (a) and (c), respectively. In addition, we follow the two outstanding works of knowledge distillation and set the temperature value to 20. If needed, we can add more experiments.
>
> Q4:Missing baselines in FS-IL and ZS-IL. Table 1 reports the results in FS and ZS settings but didn't refer to previous methods for comparison. This Github link maintains recent works in this line. https://github.com/zhoudw-zdw/Awesome-Few-Shot-Class-Incremental-Learning
>
> R4:The comment caused the rewriting of the discussion of Table 1 in the paper. Actually, our main contribution is training incrementally with neither growing the network (or any auxiliary network) nor buffering past samples since, in the real world, keeping past data may not be allowed due to privacy concerns. To put it into practice, we employ the memory recovery paradigm (MRP) to ask the learner network to remember what it has learned and meet an outstanding performance compared with the state-of-the-art works. Furthermore, we conduct an experiment (Few-Shot Incremental Learning "FS-IL" changed to Data-Allowed Incremental Learning "DA-IL") to demonstrate the additional profits of using the MRP by allowing the use of small fixed-sized memory for buffering past samples resulting in improving performance. It means our work can be employed in any memory replay-based methods to boost buffering strategy effectiveness and overcome bias problems.
>
> Q5:Missing comparison with the synthesis approaches with metadata. How well does the method perform in terms of image realism? For example, ACE uses the mean and std of the past data to synthesize images. (https://openaccess.thecvf.com/content_ICCV_2019/html/Wu_ACE_Adapting_to_Changing_Environments_for_Semantic_Segmentation_ICCV_2019_paper.html)
>
> R5:The suggestion makes good sense when you realize we distill information from the past networks to form a synthetic set of past concepts. So, the synthesis transfer set is not human natural-looking images since they are what the network has learned from the past concepts. Obviously, the quality of our synthesis images in terms of realism is not comparable with works focusing on generation realism images. Therefore, we can be satisfied when the learner network can stay learned on transfer set while learning incremented class.

---

> > ### Author Response · Authors · 2021-11-17
> > **Q6 response**
> >
> > Q6:It might be better to try the approach on more benchmarks like CIFAR-100, CUB, iNaturalist, etc.
> >
> > R6:In order to make a comprehensive comparison, we decided to put our methods against outstanding works of each category of approaches (Regularization, memory-replay, etc.). In the meantime, we follow DER [1] to reproduce the results of other works, which report all on the two datasets we also reported in the paper. However, we plan to give it a try and upload the result on CIFAR-100 before 20 Nov.
> >
> > [1] : Buzzega, Pietro, et al. "Dark experience for general continual learning: a strong, simple baseline."  NeurIPS (2020).

---

> > ### Author Response · Authors · 2021-11-22
> > **New rebuttal revised paper, More Experiments (updated)**
> >
> > We conduct a set of new experiments on CIFAR-100 in which we follow the mnemonics configuration (experimental settings, network architecture) and use their official code to evaluate and reproduce results. The evaluation results are reported in the table below, where N means the incremental phases (number of tasks). We report the DF-IL as our Data-Free Incremental Learning version in which no actual data is buffered from the past. Also, we present the DA-IL as our Data-Allowed Incremental Learning version in which a few samples (10 samples per class) are buffered from the past to be gathered with our synthesised transfer set in the incremental learning process. Moreover, we experimented with iCaRL and LUCIR  while boosting their buffer with our transfer set in the table as iCaRL+ (DA-IL) and LUCIR + (DA+IL), respectively. As shown in the table, while having the same IL method, our method is a powerful extension when compared to mnemonics.  Furthermore, it is worth noting that our method has more consistent results when increasing the number of incremental phases means our method can synthesise past samples effectively regardless of how many classes have been learned by the learner network.
> >
> > | N                 |   5   |   10  |   25  | buffer size |
> > |-------------------|:-----:|:-----:|:-----:|:-----------:|
> > | LWF               | 49.59 | 46.98 | 45.51 |      2K     |
> > | LWF + mnemonics   | 54.43 | 52.67 | 51.75 |      2K     |
> > | iCaRL             | 57.12 | 52.66 | 48.22 |      2K     |
> > | iCaRL + mnemonics | 59.88 | 57.53 | 54.30 |      2K     |
> > | BiC               | 59.36 | 54.20 | 50.00 |      2K     |
> > | BiC + mnemonics   | 60.67 | 58.11 | 55.51 |      2K     |
> > | LUCIR             | 63.17 | 60.14 | 57.54 |      2K     |
> > | LUCIR + mnemonics | 64.95 | 63.25 | 63.70 |      2K     |
> > | iCaRL + (DA+IL)   | 61.03 | 59.79 | 58.01 |      2K     |
> > | LUCIR + (DA+IL)   | 65.32 | 65.21 | 65.8  |      2K     |
> > | DF-IL (ours)      | 62.37 | 62.01 | 61.79 |      0      |
> > | DA-IL (ours)      | 63.64 | 63.18 | 63.12 |      1K     |

---

### Decision · Program_Chairs · 2022-01-20

**Decision:**

Accept (Spotlight)

**Comment:**

This paper presents a zero-shot incremental learning approach that does not store past samples for experience replay. The idea is novel and well motivated, and the paper is well written. Reviewers' comments were mainly about missing baselines, missing ablation studies, and  clarifications about the proposed method. In the revised paper, the authors provided more justifications and added new experimental results on large benchmark datasets as well as ablation studies. After discussion, all the reviewers are positive about this submission.

Thus, I recommend to accept this paper. I encourage the authors to take the review feedback into account in the final version.